# Inconel 625/AISI 413 Stainless Steel Functionally Graded Material Produced by Direct Laser Deposition

**DOI:** 10.3390/ma14195595

**Published:** 2021-09-26

**Authors:** André Alves Ferreira, Omid Emadinia, João Manuel Cruz, Ana Rosanete Reis, Manuel Fernando Vieira

**Affiliations:** 1Faculty of Engineering, University of Porto, R. Dr. Roberto Frias, 4200-465 Porto, Portugal; areis@inegi.up.pt; 2LAETA/INEGI—Institute of Science and Innovation in Mechanical and Industrial Engineering, R. Dr. Roberto Frias, 4200-465 Porto, Portugal; oemadinia@inegi.up.pt; 3SERMEC-Group, R. de Montezelo 540, 4425-348 Maia, Portugal; joaocruz@sermecgroup.pt

**Keywords:** functionally graded material, direct laser deposition, microstructure, chemical composition, hardness

## Abstract

Functionally graded material (FGM) based on Inconel 625 and AISI 431 stainless steel powders was produced by applying the direct laser deposition (DLD) process. The FGM starts with layers of Inconel 625 and ends with layers of 431 stainless steel having three intermediate zones with the composition (100-X)% Inconel 625-X% 431 stainless steel, X = 25, 50, and 75, in that order. This FGM was deposited on a 42CrMo4 steel substrate, with and without preheating. Microstructures of these FGMs were evaluated, while considering the distribution of chemical composition and grain structure. Microstructures mainly consisted of columnar grains independent of preheating condition; epitaxial growth was observed. The application of a non-preheated substrate caused the formation of planar grains in the vicinity of the substrate. In addition, hardness maps were produced. The hardness distribution across these FGMs confirmed a smooth transition between deposited layers; however, the heat-affected zone was greatly influenced by the preheating condition. This study suggests that an optimum Inconel 625/AISI 431 FGM obtained by DLD should not exceed 50% AISI 431 stainless steel.

## 1. Introduction

Functionally graded materials (FGMs) can be considered as a particular class of composites with a spatial variation of composition/microstructure along a specific direction. However, FGMs may not encompass sharp distinguishable interfaces as observed in traditional composite materials [1]. The application of FGMs can overcome challenges that exist in conventional materials and processing. It enables designers to use two complex materials that would be difficult to bond, creating compositional gradients that allow for a gradual transition between both materials without discontinuities that jeopardize the structural integrity of the component. This setting leads to fewer internal stresses and cracking, consequently improving strength [2,3].

The Functionally Graded Additive Manufacturing (FGAM) concept can be developed, i.e., the production of FGMs that are different in distribution or composition through a layer-by-layer approach [4,5]. Regarding this concept, the application of direct laser deposition (DLD), also designated by laser metal deposition (LMD), will be noticeable for depositing gradients of metals and alloys on a substrate. Densification will be obtained by solidifying consecutive melt pools generated by the laser [6]. This technique has the advantage of locally synthesizing metal/alloy gradients by mixing different powders with the desired compositions, gradually varying the mixture at intended locations [7]. However, the production of FGM components can face challenges, such as the control of mixing, melting, and cooling rate, subsequently forming intermetallic phases and cracking. The lack of bonding between tracks/layers may happen, that is, caused by unmelted particles due to using dissimilar powders that have different properties.

Regarding the DLD process, the laser/substrate relative velocity, laser scanning pattern, laser power, laser beam diameter, hatch spacing, powder feed rate, powders composition, powder gradient variation, and preheating conditions are vital parameters that must be considered [7,8]. Moreover, laser deposited materials experience complicated thermal history, presenting rapid solidification, high cooling rates, steep thermal gradients, and cyclic reheating and cooling. These conditions can produce non-equilibrium microstructures with variations layer to layer or even within individual layers. Therefore, the deposition process should be optimized while considering the characteristics of input materials [9,10].

The production of FGM by DLD has been the subject of study by several research groups. However, these products are currently limited to small samples. The construction of a component with functional gradient properties depends not only on the position of materials but also on optimizing process parameters required to control the microstructure and improve the mechanical properties in multi-material with functional gradient. High-performance and versatility FGMs can meet performance requirements and have been widely used in the fields of aerospace, biological, electromagnetic, nuclear, and photoelectric engineering [7,11].

This process uses a deposition system equipped with two or more powder feeders and can create dissimilar gradients traditionally difficult to reach. The ability to mix two or more types of powders and control the feed rate of each flow makes DLD a flexible process for manufacturing complex components for the innovative development of alloys and formation of materials with a gradient of functionality [4,12,13]. This method makes it possible to produce materials with a gradient at the microstructure level; this gradient was achieved due to the reduced and localized melting and the strong mixing movement in the melt. Thus, materials can be adapted for flexible, functional performance in particular applications. Moreover, additive manufacturing technology (AM) has surpassed the prototyping concept to produce solid components for end-users.

Regarding the production of FGM by the DLD technique, some studies mentioned the use of different systems. For the SS316/Inconel 625 system, there was an increase of mechanical and wear resistance due to the formation of secondary phases with the increase of Inconel 625 alloy content [11,14,15,16]. The increase in wear and hardness was also observed in the SS316/Inconel 718 system [17]. The FGM produced by using SS410/Inconel 625 materials demonstrated that the depositions were defected free and with good integrity along with the entire interface [18]. The effect of preheating on FGM was evaluated by using the Inconel 625/Ti6Al4V system, which was shown to promote the formation of thinner and more uniform secondary phases and free of cracks [19]. It is worth noting that there are many investigations producing FGMs by using nickel superalloys in recent years. For these alloys, a percentage increase of alloying elements, such as Cr and Mo, promotes the increase of mechanical strength and wear and corrosion resistance.

Recognizing the importance of metallic FGM and its complexity, this article explores the deposition of Inconel 625 superalloy powder gradually mixed with 431 stainless steel alloy, evaluating the influence of compositional variation, as well as preheating, on the microstructure and mechanical proprieties. The former condition was performed by preheating the substrate metal used for deposition. Although several investigations allocated the production of gradient materials by using Inconel 625 superalloy with other alloys [14,15,16], to the knowledge of the authors, the production of FGM consisting of Inconel 625 and AISI 431 has not been reported yet.

## 2. Experimental Procedure

This study included the production of compositional gradients as functionally graded material (FGM), using Inconel 625 powder (a nickel-based superalloy supplied as MetcoClad 625 by Oerlikon Metco (Westury, New York, NY, USA), so-called M625 in this study) mixed with AISI 431 stainless steel powder (a martensitic stainless steel supplied as Metco 42C from the same supplier, so-called M42C) in gradient. According to the supplier’s data sheets, M625 has a particle size range of 45–90 µm, and M42C is in a size range of 45–106 µm; the chemical composition of these alloys is presented in Table 1. Moreover, Figure 1 illustrates the morphology of these powders; the M625 particles are seen in spherical form, and M42C particles have irregular shape (non-spherical). Microscopic characterizations in this study involved a scanning electron microscopy (SEM), FEI-Quanta 400 FEG equipment FEG (ESEM, Hillsboro, OR, USA), using secondary electron (SEM/SE) and backscattered electron (SEM/BSE) imaging modes. Moreover, semi-quantitative chemical analysis was performed by using energy dispersive X-ray spectroscopy (EDX) (EDAX Genesis X4M, Oxford Instrument, Oxfordshire, UK). Structural analysis, such as crystallographic information, was performed by electron backscatter diffraction (EBSD) (EDAX-TSL OIM EBSD, Mahwah, NJ, USA) technique, applying inverse pole figure (IPF) maps.

In this study, the substrate used for deposition was 42CrMo4 steel, machined plates in 100 × 120 × 15 mm, supplied in quenched and tempered condition. This steel is widely used for manufacturing industrial components, such as gears, automotive components, and drilling joints [20,21,22]. In the current study, the production of FGM was performed on two substrates: (1) one substrate in room temperature and (2) another one preheated to 300 °C by a manual gas system. For the latter condition, the temperature was controlled by a digital thermometer, since it is essential to have a uniform temperature distribution in the substrate surface. The application of without and with preheating procedures (so-called without and with PHT in this study) aimed to evaluate the effect of cooling rate on the evolution of microstructure in deposited layers and substrate.

The consolidation of powders, required for the FGM production, was achieved by direct laser deposition (DLD) technique, using a six-axis robot KUKA KR90 R3100 model (Augsburg, Germany) connected to robotic and laser control units. This system was equipped with a laser system (LDF 3000–100), a fiber-coupled laser diode providing a wavelength range of 900–1030 nm, reaching a nominal beam power of 6000 W.

The depositions started with 100% M625 on the 42CrMo4 substrates (without PHT and PHT conditions), followed by depositing layers of 75% M625–25% M42C, 50% M625–50% M42C, 25% M625–75% M42C, and ended to 100% M42C. Feeding of powder mixtures was performed in a coaxially delivering mode for constructing compositional gradients. Moreover, argon shield gas, with 99.99% purity, was used as protection gas with a flow rate equal to 5.5 L/min to minimize contamination and oxidation of the melt pool during the DLD process. For the deposition of M625 layers, the following processing conditions were used: laser power (LP) = 2000 W, scanning speed (SS) = 6 mm/s, and feeding rate (FR) = 15 g/min. The last layers (100% M42C) were deposited with the following conditions: LP = 1500 W, SS = 10 mm/s, and FR = 15 g/min. These procedures were carried out by applying a spot size equal to 2.5 mm; the trajectory of depositions involved continuously parallel depositing, applying a 40% overlapping between tracks, followed by depositing successive layers rotated in 90° in each layer. Afterwards, printed specimens, without PHT and PHT conditions, were cooled down to room temperature. The application of these conditions was based on previous studies [23,24]. Process optimization is essential, since FGMs produced by laser deposition present microstructural variations across layers affected by different parameters, such as thermal gradients; these effects are caused by remelting and reheating cycles or cooling rate [9].

Regarding microscopic and mechanical characterizations, FGM specimens with and without PHT were prepared by using conventional metallographic techniques. Samples from each deposition were cut by using a metallographic cutoff machine with refrigeration to avoid substrate and cladding overheating. Samples were mounted in resin and polished down to 1 μm diamond suspension. However, an additional polishing step, using a 0.06 μm silica colloidal suspension mixed with ammonium hydroxide solution 25%, was needed for EBSD analysis, allowing us to obtain Kikuchi patterns [25]. The samples were taken perpendicular to the substrate surface to allow the observation of the different layers.

Similar FGM specimens were used for the microhardness test, using a fully automated DURASCAN 70 microindenter—EMCO TEST (EMCO-TEST PRÜFMASCHINEN GMBH, Kuchl, Austria). The HV hardness maps were produced by 700 indentations, applying a load of 300 g, considering 0.1 mm as the distance between the centers of every two adjacent indentations. This procedure scanned an area of 5.7 × 1.1 mm.

## 3. Results and Discussion

### 3.1. Microstructural and Chemical Evaluations

The microstructure of a FGM without PHT, from 100% M625 to 100% M42C, is illustrated in Figure 2. In this microstructure are observed some inclusions and porosities, apparently reduced by the increase in M42C alloy. However, this addition has ended up with the formation of cracks; the morphology of these defects reveal that they formed in the last layers, 100% M42C, and propagated to the layers beneath, that is, 25% M625–75% M42C.

M42C deposits are prone to cracking, and strict control of processing conditions is mandatory [23,26]. The main reason for cracking is the stresses caused by processing conditions nucleating microcracks in the brittle martensite. In this case, this was even more critical as the first layers remelted the top surface of 25% M625–75% M42C layer, and unexpected phases may have formed, increasing the brittle character of this region.

The inclusions (round black spots) are mainly complex oxides formed along the FGM, and the irregular porosities are likely caused by elemental segregation [27,28].

Figure 3 illustrates a higher magnification of the black rectangle in Figure 2, where an irregular porosity was detected. The morphology of the microstructure shown in the SEM image of Figure 3 consists of a dendritic structure embedding interdendritic regions. The elemental maps show a matrix homogeneous in Fe, Ni, and Cr and zones rich in Nb and Mo (white regions in SEM image). In regions with higher segregation, remelting occurs during the next deposition, since every layer is highly affected by the heat conducted from successive deposition, and liquation cracks formed and remained in the FGM [28,29]. This can explain why this defect decreases with an increasing amount of M42C powder.

This second phase can be the Laves phase, resulting from the microstructural segregation of Nb and Mo elements from the liquid due to rapid solidification during deposition. The formation of the Laves phase, or even of carbides, in the austenitic matrix has been observed in several studies [30,31,32,33,34]. Its presence was also revealed by microscopic observations and EDS analysis in a similar study on laser cladding of Inconel 625 alloy [24]. The amount of the Laves phase can be reduced by post-deposition heat treatments that homogenize the material by reducing chemical composition gradients [35].

The formation of secondary phases in FGM depends on the processing history [36], being possible to minimize the proportion of Laves phase in the microstructure by preheating the substrate [24]. The application of PHT reduces the cooling rate of the deposited material, allowing the diffusion of Nb and Mo elements in the matrix, and thus reducing the amount of the interdentritic Laves phase. The effect of the interdentritic Laves phase on the material hardness is not consensual. It has been reported that phase Laves can either increase hardness [32] or decrease it [37], in this case, due to the reduction of carbides in the matrix as a consequence of the Nb and Mo segregation to the interdendritic regions.

Figure 4 illustrates several details of the FGM microstructure. The images show that the microstructure is predominantly composed of columnar dendrites grains, a characteristic of laser-deposited structures [38,39]. This structure is formed since the thermal gradient and the solidification rate favor columnar–dendritic solidification morphology grains. There are two narrow zones of the cladding where planar and equiaxed grains morphologies can occur; a planar interface zone forms at the interface with the substrate due to the very high thermal gradient, and equiaxed morphology can be observed near the surface of the melt pool, resulting from the decreasing thermal gradient as the cladding solidifies. Typically, in this process, columnar grains grow parallel to the main heat flow across the material being solidified.

During the deposition of several layers of the same material, each new layer remelts the surface of the last deposited, replacing the zone of equiaxed grains with columnar ones. As a result, the equiaxed region is limited to the upper surface of the cladding. However, Figure 4(A4,B4) evidenced some equiaxed grains appearing inside the FGM, mainly in the upper region of the 50% M625 + 50% M42C zone. This effect can be explained by the composition of the liquid formed and the difficulty in solute redistribution, which can cause the appearance of equiaxed morphology, as reported in other studies [33,40].

SEM images of Figure 4(A1,B1) reveal a dilution zone resulting from the melting of the substrate during laser processing and ensuring the bonding between the cladding and substrate. Moreover, well-bonded layers are seen all across the FGM (Figure 4(A2–6,B2–6)). The remelting of the upper region of the last deposited layer and the mixing with melted powders ensure the bonding between these layers and a continuous chemical composition gradient across the entire FGM, as shown in Figure 5.

The presence of Fe from the substrate in the first M625 layers is more pronounced in the PHT condition. This difference is caused by the thermal energy having caused a higher dilution of the preheated substrate, with more Fe incorporating the melt pool, as observed in other studies [23]. As shown in Figure 5, the fluctuation of the Fe concentration in the PHT condition implies the depletion of the Ni, Cr, and Mo. However, Fe from the substrate melt depletes at about 1.5 mm of the FGM regardless of preheating conditions; afterward, the Fe concentration increases with increasing M42C powder. Regarding other elements, the Cr distribution seems constant throughout the FGM; this homogeneity results from this element existing in both M625 and M42C powders in similar amounts. Some fluctuations in Nb and Mo profiles are stronger for the PHT condition in layers close to the substrate, up to about 1.5 mm; as expected, concentrations of these elements decrease with increasing M42C percentage. This increment in steel powder is also associated with a decrease in Ni.

Figure 6A,B reveals an influence of PHT on the microstructure of the first M625 layers, i.e., in the substrate vicinity. Without PHT, a layer with almost 50 µm of planar grains was formed, while with PHT, only columnar structures are observed. As already mentioned, this zone of planar grains is formed due to the very high thermal gradient in the contact zone of the melt pool with the cold substrate; PHT significantly reduces this gradient, and solidification conditions lead to the formation of columnar structures. These observations are consistent with similar studies [24,41]. This layer with planar grains has been interrupted by proceeding the solidification; that means that the solid–liquid interface growth rate and thermal gradient in the melt pool changed in favor of columnar-dendritic growth.

The microstructural evolution in the FGM was evaluated in detail through localized chemical analysis, using EDS. The EDS analysis of the zones illustrated in Figure 6A,B and presented in Table 2 also confirmed that preheating caused the increase of Fe in the 100% M625 layers of the FGM (zones Z1 and Z3), and strongly promoted the diffusion of alloying elements of the M625 layer into the substrate with PHT (zones Z2 and Z4). This diffusion of Ni, Mo, Nb, and Cr into the preheated substrate, associated with the depletion of Fe, can affect the mechanical properties, such as hardness, of the substrate in the diffused zone.

Figure 6C,D gives more details about the formation of secondary phases in FGMs. The EDS analysis of the round dark zones, identified as zones Z5 and Z8, are complex oxides with a composition (Cr, Ni, Fe, Nb, Mo, Mn, Si)_x_O_y_. The microstructures also reveal the presence of lighter (white and gray) regions. The results of Table 2 confirm that these regions are mainly Laves phase and carbides. A comparison of the chemical composition of the zones indicated in Figure 6C (Z6 and Z7) and Figure 6D (Z9) shows that preheating affects their composition by increasing the iron content and decreasing the nickel content, in accordance with Figure 5.

The PHT effect on the segregation for the interdendritic zones of Nb and Mo elements, which are the main compositional elements of the Laves phase, is not apparent throughout the FGM. However, close to the interface, this variation seems evident due to the decrease of the Laves phase by the PHT effect, as observed by comparing the representations of zone Z1 in Figure 6A,B. This effect is, in part, explained by the increase in Fe content in the cladding. Furthermore, the volume fraction of the Laves phase depends on the alloy solidification process, and higher cooling rates in this region, typical of the cladding without PHT, reduce the time for Nb and Mo diffusion and lead to their accumulation in the interdendritic spaces.

Typically, in the DLD process, the interface and heat-affected zone (HAZ) are critical regions. In fact, the heat input in these regions is much smaller than in conventional welding processes due to the localized molten region created by the laser. Consequently, the cooling rate is very high at the beginning of cladding solidification, promoting a significant microstructural change in the HAZ. This change can increase the hardness and decrease the toughness in the substrate HAZ.

In this study, preheating the substrate to a temperature of 300 °C promoted not only a microstructural change at the interface, inhibiting the formation of the planar grain layer (Figure 6), but also in the HAZ, causing the formation of coarser structures and reducing the formation of martensite, as shown in Figure 7. Thus, PHT leads to a microstructure that can reduce crack initiation conditions during the in-service use of the coated steel. Figure 7 also showed a more intense diffusion at the interface of the FGM produced with a pre-heated substrate, with the mutual interpenetration of the substrate and cladding leading to a diffuse interface. The analysis of Table 2 confirms this microstructural observation on the effect of PHT on diffusion; a higher percentage of iron in the cladding (as evidenced by comparing Z1 and Z3) and a higher percentage of Nb, Mo, Cr, and Ni from the cladding, in the substrate (as highlighted by comparing Z2 and Z4) was detected.

Previous studies confirmed that PHT positively influenced the microstructure and the mechanical properties in substrates processed by DLD [42]. In addition, it promoted a reduction of residual stresses of about 40%, as well as the reduction and attenuation of distortions [43], permitting a better distribution of stresses between the cladding and the substrate, as well as preventing the formation of intermetallic phases (as secondary phases), decreasing hardness, and improving mechanical properties.

An EBSD analysis was performed to observe the morphology and grain distribution of the FGMs, as illustrated in Figure 8. As expected, considering SEM/BSE images of Figure 6A,B, there is a layer with smaller and equiaxed grains in the vicinity of the substrate. This smaller grain size is more evident in the sample without PHT, due to the influence of the cold substrate. However, the microstructure in both FGMs is mainly composed of columnar grains that grow perpendicular to the substrate, i.e., in the direction of deposition and heat flow. The growth of columnar–dendritic structures along the deposition direction occurs when the temperature gradient component in that direction is larger than other temperature components in the melt pool [44,45].

In the layers deposited with the 50% M625 + 50% M42C powder mixture, there is a zone with equiaxed grains, probably formed by the complex chemical composition and the heat accumulation, which induced a partial reduction of the high thermal gradient. However, this localized microstructural alteration is again replaced by columnar grains, not being maintained until the last deposited layers, contrary to what has been seen in other studies [41]. It should also be noted that the size of columnar grains decreases as more layers are deposited.

For the first two compositions (100% M625 and 75% M625 + 25% M42C), the grains of the FGM without PHT (Figure 8A) are thicker and longer than those of the FGM with PHT (Figure 8B), which shows another effect of reducing the thermal gradient by the application of PHT.

EBSD images also show that some grains form in one region and spread to the next, with different compositions. This indicates epitaxial growth in successive layers. This type of growth, which favors the bonding between layers, occurs because the deposition of a new layer remelts the surface of the previous one. This remelting/solidification process allows the grains from the previous deposition to act as nucleation sites for the solidification of new grains.

The images in Figure 8 do not show the formation of a preferential orientation in the microstructure, since no color is dominant in these inverse-pole figures.

### 3.2. Microhardness Mapping

In this study, the composition gradient from the substrates to the upper layers, with a continuous increase in the amount of martensitic steel, should show an evolution of hardness along with the deposited layers. In fact, previous studies on the deposition of monolayers of these materials indicate average hardness values greater than 500 HV for M42C [23] and approximately 250 HV for M625 [24]. However, no marked variation in hardness was measured across the FGMs, as illustrated in the microhardness maps shown in Figure 9. The figure also shows no significant differences in FGMs processed with and without PHT, which proves that the influence of PHT on the microstructure is not very significant, except for the planar morphology of the first deposited layers.

The relatively low hardness of the M42C-rich layers is explained by a slower cooling rate in these layers, which are the last to be deposited, inhibiting an extensive martensitic transformation, and by these layers having about 10 wt.% Ni, which, being one austenite stabilizer, also hinders the martensitic transformation. Finally, except for the last layer, all others undergo a self-tempering process of the martensite that may have formed.

Figure 9 reveals that the hardest zone obtained is in the heat-affected zone (HAZ) of the FGM produced without PHT, meaning that preheating application promoted a reduction in the cooling rate in the substrate, reducing the formation of martensite in this zone, as already discussed.

The higher hardness of the FGM with PHT (indicated by a red arrow) was measured in the M625 region, which can be attributed to compositional fluctuations leading to a local concentration of hard Laves phase/carbides.

This evolution of hardness shows that, up to 50% of M42C powder, which is significantly less expensive, can be added to M625 powder without inducing significant changes in hardness and microstructure, as discussed above. Larger amounts of M42C should not be added, as they can lead to cracking.

## 4. Conclusions

In this study, the production of functionally graded material (FGM) by direct laser deposition (DLD) technique was evaluated. The deposition started with layers of nickel-based superalloy (M625 powders) and ended with layers of martensitic stainless steel (M42C powders). Three mixtures of powders were used in intermediate deposits, sequentially increasing by 25 wt.% the amounts of M42C powder. Moreover, the influence of preheating the 42CrMo4 steel substrate on the microstructural and hardness evolution in FGMs were evaluated. The main conclusions of this study are as follows:Cracking-free production of the Inconel 625/AISI 431 steel FGM, applying DLD, is only verified up to a certain composition. The addition of stainless steel cannot exceed 50 wt.%.The metallurgical bonding of deposits to substrates and between the various layers of the FGM is ensured by the diffusion in the liquid state of the alloy constituents, the remelting effect, and epitaxial growth.The grain microstructure in Inconel 625/AISI 431 FGM is essentially columnar, regardless of preheating.Preheating influenced the microstructural evolution and microhardness in the substrate and the first deposited layers; the region of planar grains observed in the vicinity of the substrate only formed without preheating. A marked increase in grain size and a reduction in martensite was observed in the preheated substrate HAZ, decreasing the hardness of this region.

## Figures and Tables

**Figure 1 materials-14-05595-f001:**
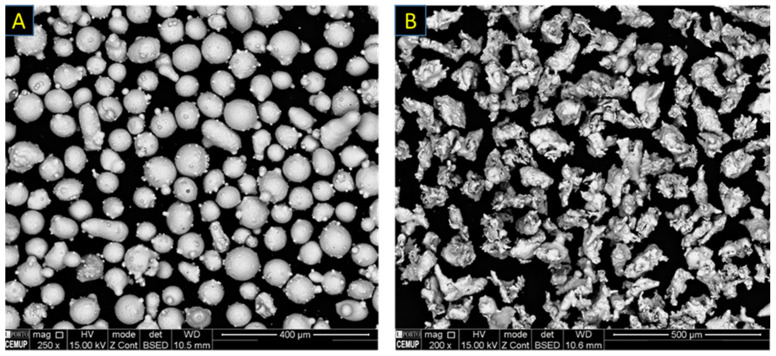
Morphology of (**A**) M625 and (**B**) M42C powders illustrated by SEM/BSE technique.

**Figure 2 materials-14-05595-f002:**
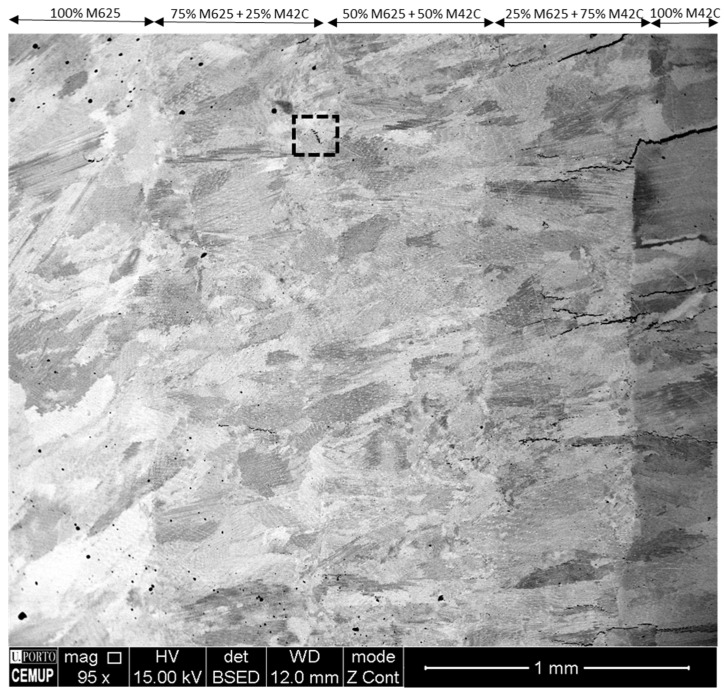
Microstructural evolution across the longitudinal section of the FGM specimen without PHT, SEM/BSE image.

**Figure 3 materials-14-05595-f003:**
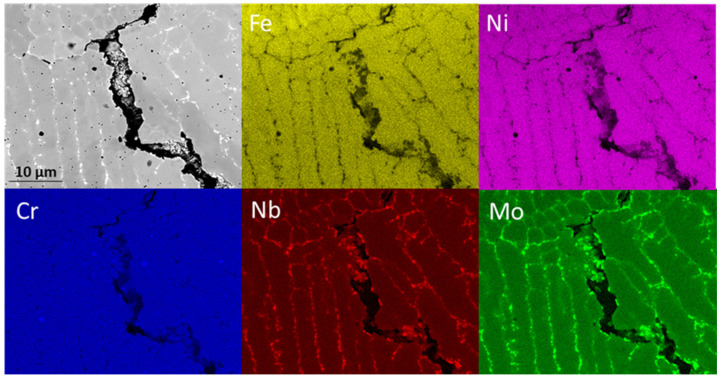
SEM image and EDS elemental maps of a discontinuity observed in 75% M625 + 50% M42C layer, illustrated as black square inset in Figure 2.

**Figure 4 materials-14-05595-f004:**
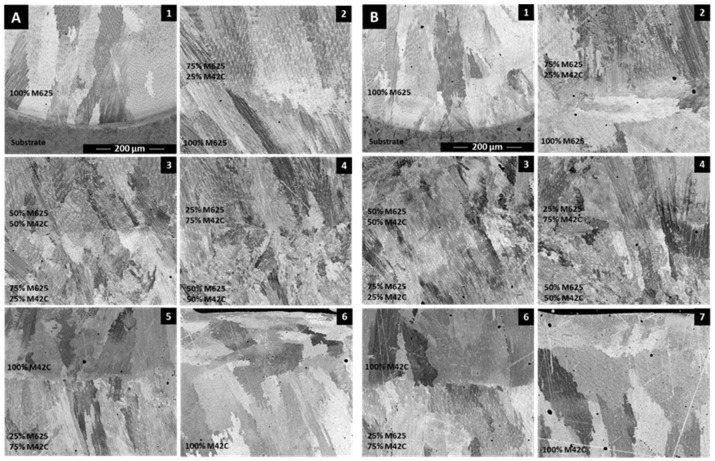
Microstructural evolution in the transversal cross-section of FGMs produced (**A**) without and (**B**) with the application of PHT (SEM/BSE images).

**Figure 5 materials-14-05595-f005:**
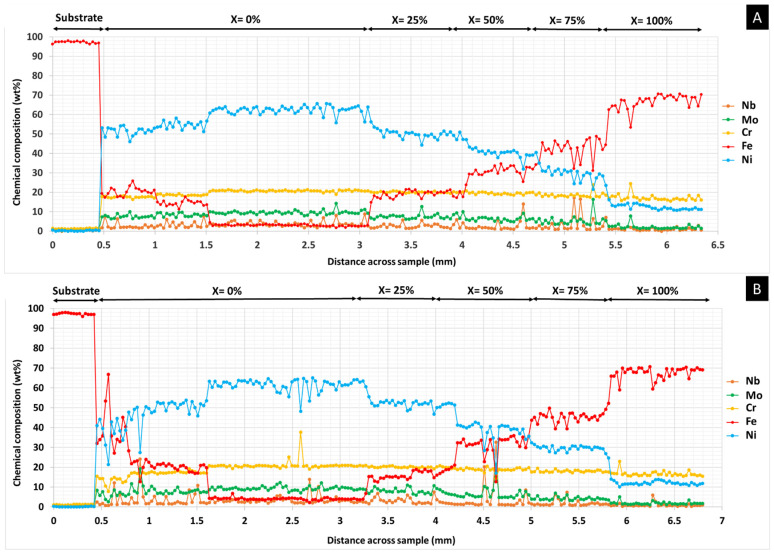
FGM linear chemical analysis—(**A**) without PHT and (**B**) with PHT. The composition of each FGM zone is (100-X)% M625 + X% M42C.

**Figure 6 materials-14-05595-f006:**
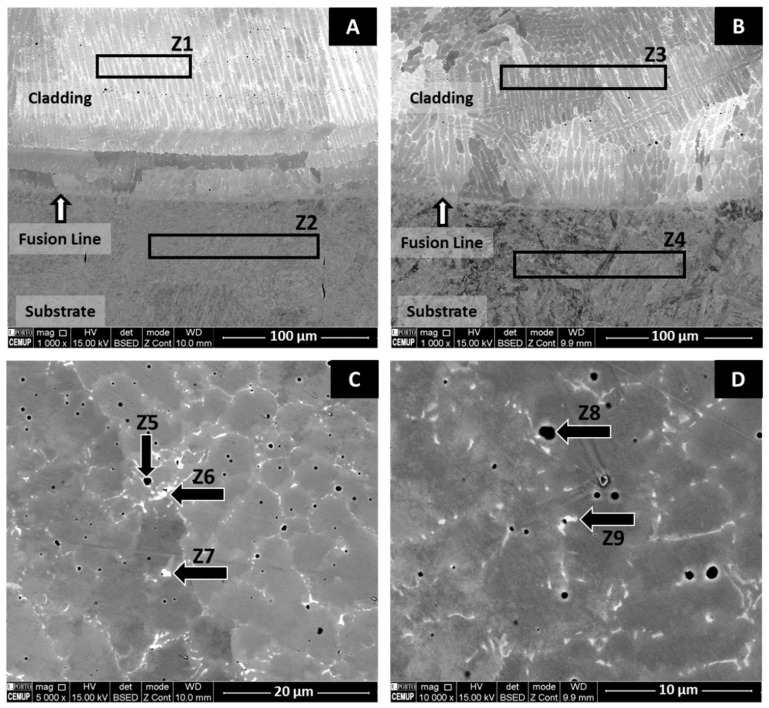
SEM/BSE images showing (**A**,**B**) the FGM/substrate interface and (**C**,**D**) higher magnification images for secondary phases analysis from the 50% M625 + 50% M42C layers. (**A**,**C**) are from FGM produced without PHT, and (**B**,**D**) with PHT.

**Figure 7 materials-14-05595-f007:**
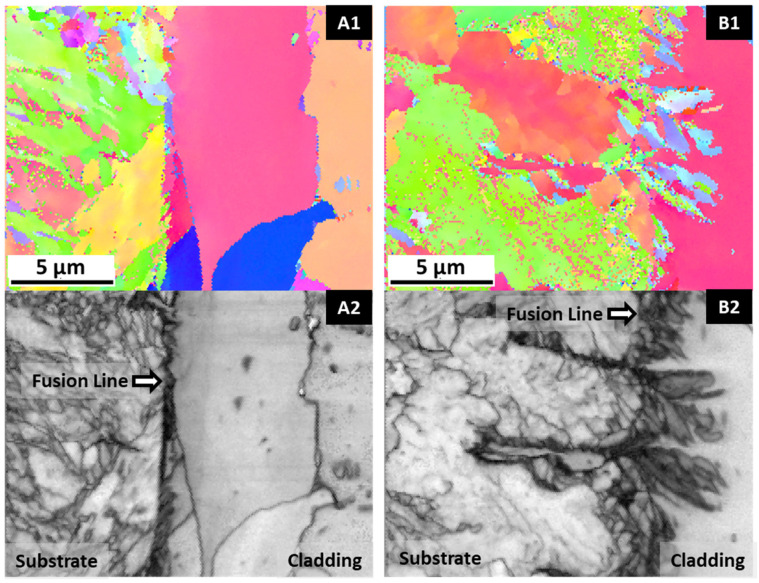
Morphology of the cladding/substrate interface with substrate and HAZ observed by EBSD technique: (**A1**,**A2**) without PHT and (**B1**,**B2**) with PHT, showing grain maps and SEM images, respectively.

**Figure 8 materials-14-05595-f008:**
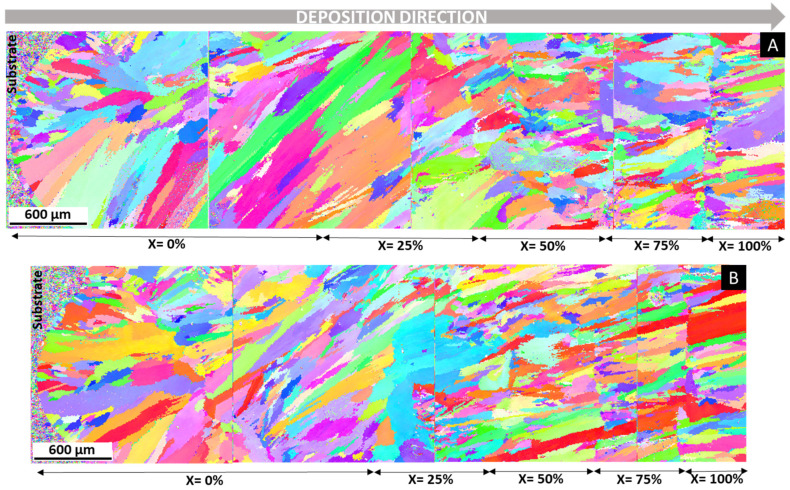
EBSD inverse-pole-figure (IPF) map of the cross-section of FGMs (**A**) without PHT and (**B**) with PHT, showing the morphology and orientation of grains. The composition of each FGM zone is (100-X)% M625 + X% M42C.

**Figure 9 materials-14-05595-f009:**
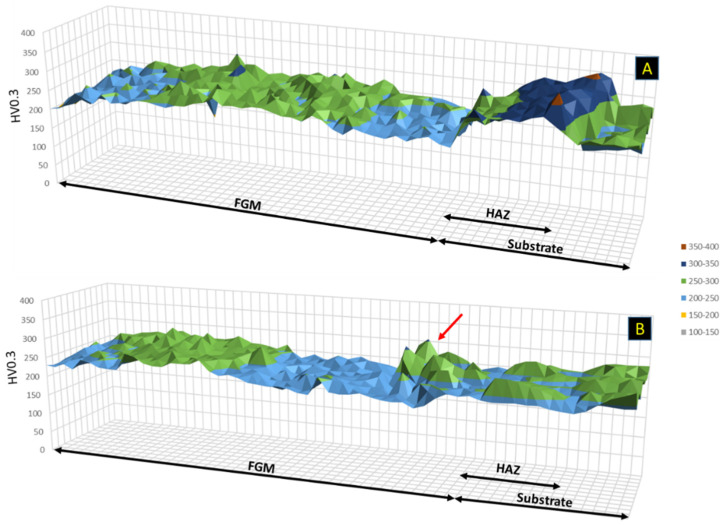
FGM microhardness mapping. (**A**) without PHT and (**B**) with PHT.

**Table 1 materials-14-05595-t001:** Chemical composition (wt.%) of the FGM powder alloys used in this study.

Powders	Fe	Ni	Cr	Mo	Nb	Si	C
M42C	78.6	1.9	17.3	-	-	2.0	0.2
M625	4.1	60.8	21.3	9.2	4.6	-	-

**Table 2 materials-14-05595-t002:** EDS analysis (wt.%) performed on the FGM zones illustrated in Figure 6. Z1, Z2, Z5, Z6, and Z7 are from FGMs without PHT, and Z3, Z4, Z8, and Z9 from FGMs with PHT.

Zone	C	O	Si	Nb	Mo	Cr	Fe	Ni	Mn
**Z1**	0.9	-	0.5	2.9	7.8	16.6	22.3	49.0	-
**Z2**	0.9	-	0.3	0.0	0.0	1.5	97.3	0.0	-
**Z3**	0.7	-	0.5	2.7	6.2	13.6	37.1	39.2	-
**Z4**	0.7	-	0.4	1.2	3.2	6.9	69.1	18.5	-
**Z5**	0.7	16.9	0.4	9.2	5.2	28.2	8.8	22.5	8.1
**Z6**	1.8	-	2.3	12.0	24.4	14.3	10.5	34.7	-
**Z7**	0.7	-	2.4	11.8	24.3	15.3	10.0	35.5	-
**Z8**	1.1	12.9	8.3	6.3	6.4	19.9	12.0	27.5	5.6
**Z9**	1.0	-	1.8	5.2	26.2	16.0	13.9	35.9	-

## Data Availability

Not applicable to this study.

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
