# Peer review of "Inconel 625/AISI 413 Stainless Steel Functionally Graded Material Produced by Direct Laser Deposition"

_materials, 2021, doi:10.3390/ma14195595_

Round 1

Reviewer 1 Report

Please find in attach some observations and or recommandations for your contribution. 

Author Response

Response to Reviewer 1 Comments

Manuscript number: metals-1371436

Title: "Inconel 625 / AISI 413 Stainless Steel Functionally Graded Material Produced by Direct Laser Deposition”

Authors: André Alves Ferreira, Omid Emadinia, João Manuel Cruz, Ana Rosanete Reis, Manuel Fernando Vieira

Dear reviewer, thank you for your comments and the positive evaluation of our work. Below is the response to your comment. The changes to the manuscript referred to are highlighted in green.

Point 1: Line 92 - For the seek of the homogeinity please use AISI 413 and not SS431

Response 1: Thanks for the comment. We corrected this information in this revised version.

Point 2: Line 118 - Please indicate how the temperature is maintained constant during the whole fabrication.

Response 2: The substrate is pre-heated before starting the depositions. Substrate heating is not ensured during complete FGM production because, after the first depositions, the most important factor becomes the temperature of the layers that have just been deposited.

Point 3: Lines 132-133 - the temperature in the melt pool. You would mention the temperature at the surface of in the melt pool.

Response 3: You're right, the system controls the surface of the melt pool. However, our latest measurements lead us to question the accuracy of this control and we prefer to remove this sentence.

Point 4: Would you please the direction for observation ‘parallel to disposition axis or what ever… Also, please indicate the etch solution and etching conditions.

Response 4: We have added the required information to the manuscript. However, we must emphasize that the samples were not etched; a light etching was obtained by adding NH4OH to the colloidal silica solution during the last polishing step.

Point 5: Line 166 - It seems that the microstructure corresponds to preheating free condition. Please indicate the pre-heating condition for the seek of the clarity.

Response 5: We have inserted the information in the revised version of the manuscript.

Point 6: Line 171 - M42C deposits are prone to INTERDENRITIC ( ??) cracking.

Response 6: References refer to macro-cracks not referring to the origin of the cracks.

Point 7: Line 177 - Would you please add for example a table (or just the value) indicating the coefficients of thermal expansion of both M625 and M42C. Have you data about coefficients of thermal expansion of X% M625+(1- X%) M42C

Response 7: We appreciate the reviewer's comment that allows us to correct information that may mislead the reader. In fact, Inconel's coefficient of thermal expansion is higher than that of AISI 413 steel (no information is available concerning FGM layers), but cracks appear in this material because its martensitic structure is more fragile and unable to accommodate, by multiplying dislocations, the residual stresses.

Point 8: Line 204- thus reducing the 204 amount of INTERDENDRITIC ??? Laves phase. Please better emphasize this part.

Response 8: We have inserted the information in the manuscript.

Point 9: Line 205- This not a clear statement. The effect of Laves phase in the FORMATION ?? of the dendritic structure… Or The effect of INTERDENDRITIC Laves phase on .. ; the hardness ?

Response 9: We have corrected the revised version according to reviewer's suggestion.

Point 10: Line 208- Consequence of Nb and Mo segregation. Please emphasize whether the segregation is interdendritic regions or not.

Response 10: According to the reviewer's indications, the text has been improved by emphasizing the segregation of the chemical elements to interdendritic regions.

Point 11: Line 210- Please emphases if: the microstructure is predominantly composed of columnar DENDRITIC GRAINS, the grains that are columnar and not dendrite

Response 11: The text was modified to incorporate the reviewer's suggestion.

Point 12: Line 212- Idem; rate favour columnar-dendritic solidification morphology GRAINS. These are the grains that are columnar and not dendrite.

Response 12: Also at this point, the text was modified to incorporate the reviewer's suggestion.

Point 13: Line 212- where planar and equiaxed GRAINS morphologies; for a seek the of clarity: planar INTERFACE

Response 13: We have corrected the revised version according to reviewer's suggestion.

Point 14: Please increase the size of letter showing the curve

Response 14: As indicated by the reviewer, the font size has been increased throughout Figure 5.

Point 15: Line 250- increasing M42C percentage

Response 15: The text was modified to incorporate the reviewer's suggestion.

Point 16: Please indicate the distance of these zones from the substrate (as reference line). Please add a column in table 2 and for indicating if the zone is with PHT or without.

Response 16: We add to Figure 6 the indication of the substrate and the cladding regions. Instead of adding a new column, we add the requested information in the caption of Table 2..

Point 17: Line 282- This effect is, in part, is explained……

Response 17: As indicated by the reviewer, we have made the indicated changes.

Point 18: Line 297- It would be better to use crack initiation

Response 18: As indicated by the reviewer, we have made the indicated changes.

Point 19: Line 298- Please be more clear: more intense diffusion of which element ? All ?

Response 19: According to the reviewer's indications, a text was added clarifying this aspect.

Point 20: Figure 7 2B- In SEM graph, it seems there exist some incurved lines. These are the artefact or a specific feature of microstructure

Response 20: The lines observed are specific microstructure characteristics.

Point 21: In figure 8 and all other figure- Could you please add in your photos the direction of fabrication and also indicate what was the orientation for cutting the samples and their cross-sectioning

Response 21: We have added information that allows the identification of the various zones (we emphasize that the substrate region is also identified in the images).

Reviewer 2 Report

  1. Experimental Procedure 

The Experimental Procedure part mainly focuses on the objective description of experimental steps and material parameters. It seems that the evaluation of material properties should be reduced, such as: “This steel is a low alloy structural steel, presenting high strength and toughness, with good fatigue behavior and machinability. Thus, it is widely used for manufacturing industrial components such as gears, automotive components, and drilling joints”. 

  1. Results and discussion 

2.1 The author stated: “The main reason for cracking is the stresses caused by processing conditions, namely the different coefficients of thermal expansion, nucleating microcracks in the brittle martensite”. I found that the internal stress of the material caused by the different expansion coefficient is affected by the position. For example, in Figure 2, the crack propagation of the 25%M625-100%M42C transition layer suddenly stops or turns. So, what is the difference between the crack propagation caused by the internal stress of the 25% M625-75% M42C layer and the crack propagation caused by the internal stress of the 25% M625-100% M42C transition layer? 

2.2 For the 75% M625+50% M42C transition layer, the author stated: “Figure 3 illustrates a higher magnification of the black rectangle in Figure 2 where an irregular porosity was detected”. However, I found that there is also irregular porosity in the 75% M625+25% M42C layer (Figure 2 A horizontal crack is found at one-third of the position from bottom to top). Please explain why. 

Author Response

Response to Reviewer 2 Comments

Manuscript number: metals-1371436

Title: “Inconel 625 / AISI 413 Stainless Steel Functionally Graded Material Produced by Direct Laser Deposition”

Authors: André Alves Ferreira, Omid Emadinia, João Manuel Cruz, Ana Rosanete Reis, Manuel Fernando Vieira

Dear reviewer, thank you for your comments and the positive evaluation of our work. Below is the response to your comment. The changes to the manuscript referred to are highlighted in blue.

Point 1: The Experimental Procedure part mainly focuses on the objective description of experimental steps and material parameters. It seems that the evaluation of material properties should be reduced, such as: “This steel is a low alloy structural steel, presenting high strength and toughness, with good fatigue behavior and machinability. Thus, it is widely used for manufacturing industrial components such as gears, automotive components, and drilling joints”.

Response 1: Thanks for the comment. As suggested, we have significantly reduced this part of the text by highlighting only the industrial importance of this substrate.

Point 2: 2.1 The author stated: “The main reason for cracking is the stresses caused by processing conditions, namely the different coefficients of thermal expansion, nucleating microcracks in the brittle martensite”. I found that the internal stress of the material caused by the different expansion coefficient is affected by the position. For example, in Figure 2, the crack propagation of the 25%M625-100%M42C transition layer suddenly stops or turns. So, what is the difference between the crack propagation caused by the internal stress of the 25% M625-75% M42C layer and the crack propagation caused by the internal stress of the 25% M625-100% M42C transition layer?

Response 2: It may not have been explained well in our manuscript, but we consider that we are talking about two different types of defects. Those observed for the layers with about 75% M625 + 25% M42C (the rectangle we have pointed out and those well-referred to by the reviewer) are pores that form due to interdendritic segregation and solidification shrinkage; the defects that seem to start in the layer consisting only of M42C steel and propagate to the interior of the clad, on the other hand, are cracks resulting from the processing and enhanced by the martensitic structure of each layer, which the subsequent deposition will only later temper.

Point 3: For the 75% M625+50% M42C transition layer, the author stated: “Figure 3 illustrates a higher magnification of the black rectangle in Figure 2 where an irregular porosity was detected”. However, I found that there is also irregular porosity in the 75% M625+25% M42C layer (Figure 2 A horizontal crack is found at one-third of the position from bottom to top). Please explain why.

Response 3: As mentioned in the previous answer, these porosities result from interdendritic segregation and solidification shrinkage being, as the reviewer mentions, more frequent for 75% M625 + 25% M42C. We do not indicate in which layer the defect in Figure 3 forms because it appears to us to be in the transition between 75% M625 + 25% M42C and 50% M625 + 50% M42C. Apparently, microsegregation problems are intensified when 25% steel powder is added to nickel superalloy powders.
